# TMEM266 is a functional voltage sensor regulated by extracellular Zn$^{2+}$

**Ferenc Papp[1,2], Suvendu Lomash[1], Orsolya Szilagyi[1], Erika Babikow[1], Jaime Smith[1], Tsg-Hui Chang[1], Maria Isabel Bahamonde[1], Gilman Ewan Stephen Toombes[1], Kenton Jon Swartz[1]\***

[1]Molecular Physiology and Biophysics Section, Porter Neuroscience Research Center, National Institute of Neurological Disorders and Stroke, National Institutes of Health, Bethesda, United States; [2]MTA-DE-NAP B Ion Channel Structure-Function Research Group, Department of Biophysics and Cell Biology, Faculty of Medicine, University of Debrecen, Debrecen, Hungary

**Abstract** Voltage-activated ion channels contain S1-S4 domains that sense membrane voltage and control opening of ion-selective pores, a mechanism that is crucial for electrical signaling. Related S1-S4 domains have been identified in voltage-sensitive phosphatases and voltage-activated proton channels, both of which lack associated pore domains. hTMEM266 is a protein of unknown function that is predicted to contain an S1-S4 domain, along with partially structured cytoplasmic termini. Here we show that hTMEM266 forms oligomers, undergoes both rapid (μs) and slow (ms) structural rearrangements in response to changes in voltage, and contains a Zn$^{2+}$ binding site that can regulate the slow conformational transition. Our results demonstrate that the S1-S4 domain in hTMEM266 is a functional voltage sensor, motivating future studies to identify cellular processes that may be regulated by the protein. The ability of hTMEM266 to respond to voltage on the μs timescale may be advantageous for designing new genetically encoded voltage indicators.

DOI: https://doi.org/10.7554/eLife.42372.001

**\*For correspondence:**
swartzk@ninds.nih.gov

## Introduction

Electrical signaling is a widespread mechanism of cellular communication that is essential for multi-cellular organisms. In eukaryotes, the orchestrated activity of voltage-activated cation channels gives rise to action potentials that are used to trigger both excitation-secretion and excitation-contraction coupling (*Hille, 2001*). The voltage-activated K$^+$ (Kv), Na$^+$ (Nav) and Ca$^{2+}$ (Cav) channels that are required for these forms of electrical signaling all contain S1-S4 domains capable of sensing changes in membrane voltage and triggering opening or closing of a coupled ion selective pore (*Swartz, 2008*). A growing number of X-ray and cryo-EM structures have provided a foundation for understanding the molecular mechanisms of voltage sensing in voltage-activated ion channels (*Hite and MacKinnon, 2017*; *Hite et al., 2017*; *Long et al., 2007*; *Payandeh et al., 2012*; *Payandeh et al., 2011*; *Shen et al., 2017*; *Sun and MacKinnon, 2017*; *Tao et al., 2017*; *Wang and MacKinnon, 2017*; *Whicher and MacKinnon, 2016*; *Wu et al., 2016*; *Wu et al., 2015*; *Yan et al., 2017*; *Zhang et al., 2012*), and it has been established that movements of the Arg-rich S4 helices drive voltage-sensor activation (*Bezanilla, 2008*; *Swartz, 2008*). S1-S4 voltage-sensing domains have also been identified in voltage-sensitive phosphatases (VSPs), where membrane voltage can regulate the catalytic activity of a cytoplasmic lipid phosphatase domain (*Grimm and Isacoff, 2016*; *Kohout et al., 2010*; *Kohout et al., 2008*; *Li et al., 2014*; *Murata et al., 2005*; *Okamura et al., 2009*; *Sakata et al., 2016*). Subsequent studies showed that voltage-activated proton (Hv1) channels

contain an S1-S4 domain that also forms the permeation pathway for protons (*Berger and Isacoff, 2011*; *Lee et al., 2009*; *Mony et al., 2015*; *Ramsey et al., 2006*; *Sasaki et al., 2006*).

In the present study, we investigated hTMEM266 (formally known as C15orf27), a membrane protein of unknown function that contains an identifiable S1-S4 domain. Although the sequence of the S1-S4 domain of hTMEM266 is similar to Hv1, it was previously reported to not form functional voltage-activated proton channels (*Kim et al., 2014*; *Musset et al., 2011*). Here we explore whether the S1-S4 domain in hTMEM266 can sense changes in membrane voltage. We demonstrate that its S4 helix can support voltage-dependent opening when transplanted into either Hv1 or Shaker Kv channels. Using voltage-clamp fluorimetry, we establish that hTMEM266 undergoes distinct conformational rearrangements in response to changes in membrane voltage, and we show that extracellular $Zn^{2+}$ ions regulate these conformational dynamics. We conclude that hTMEM266 contains a functional voltage sensor with properties that distinguish it from conventional S1-S4 voltage-sensing domains.

## Results

### Sequence and architecture of hTMEM266

The hTMEM266 protein can be identified in BLAST searches using the Hv1 channel as a query (*Kim et al., 2014*; *Musset et al., 2011*). The hTMEM266 sequence is 531 amino acids long and is predicted to contain four transmembrane (TM) helices that are flanked by an N-terminus containing approximately 100 residues and a C-terminus containing about 310 residues. The four TMs are clearly related to the S1-S4 voltage-sensing domains found in Hv1, VSPs and voltage-activated cation channels (*Figure 1A* and see below). Among proteins that contain S1-S4 voltage-sensing domains, TMEM266 has the most restricted distribution among species, with orthologous sequences restricted to the genomes of vertebrates (*Figure 1—figure supplement 1*). The sequences of TMEM266 orthologs are well conserved, with greater than 50% identity among the most distant branches of the tree and over 80% sequence identity within the S1-S4 domain (*Figure 1—figure supplement 2*). We searched for sequences homologous to either the N- or the C-termini of hTMEM266 using programs designed to identify distant sequence relationships in proteins (See Materials and methods) but we could not identify other protein sequences with similarity to these regions. The sole region where there is homology with other proteins is the S1-S4 domain, where Hv1 is the closest relative.

To explore the architecture of hTMEM266, we constructed a homology model (*Figure 1B*) using the Phyre2 server and available crystal structures of the Hv1 S1-S4 domain (*Takeshita et al., 2014*) (PDB ID: 3WKV), as well as the Hv1 coiled-coil domain (*Fujiwara et al., 2013*) (PDB IDs: 3A2A and 3VMX). (Note that the Hv1 structure 3WKV is of a chimeric construct, and thus our homology model should be interpreted cautiously.) Secondary structure predictions suggest that the N-terminus of hTMEM266 contains two short helices that may form a compact domain and that the initial part of the C-terminus forms a helical extension of the S4 helix that is likely to form a coiled-coil (*Figure 1B*), similar to the C-terminal coiled-coil domain of Hv1 that serves to assemble Hv1 into dimers (*Koch et al., 2008*; *Lee et al., 2008*; *Qiu et al., 2013*; *Tombola et al., 2008*; *Tombola et al., 2010*). Whereas the Hv1 sequence terminates at the coiled-coil, hTMEM266 family members have ~250 additional residues following the coiled-coil that is predicted by multiple secondary structure prediction programs to be structurally disordered (*Figure 1B*). (See Materials and methods).

The S1-S4 domain of hTMEM266 has key features that are common to other S1-S4 voltage-sensing domains. Like Hv1, the S4 helix of hTMEM266 contains three Arg residues (R208, R211 and R214) positioned according to the RXXRXXR motif common to the S4 helix of all voltage-sensing domains, where X is typically a hydrophobic residue. The S1-S4 domain of hTMEM266 also has a triad of residues (F149 and E152 in the S2 helix, as well as D174 in the S3 helix) that in other S1-S4 domains has been termed the 'charge-transfer center' because they facilitate movement of the positively charged residues in S4 across the membrane (*Tao et al., 2010*). In the homology model, all three Arg residues on the S4 helix face in towards the domain core with the middle Arg positioned closest to F149 within the charge transfer center, leaving the first Arg external to F149, and the third Arg internal to F149 (*Figure 1C*). Like other S1-S4 domains, hTMEM266 also contains acidic and/or polar residues positioned above (E118 on the S1 helix and S142 on the S2 helix) and below (E152 on

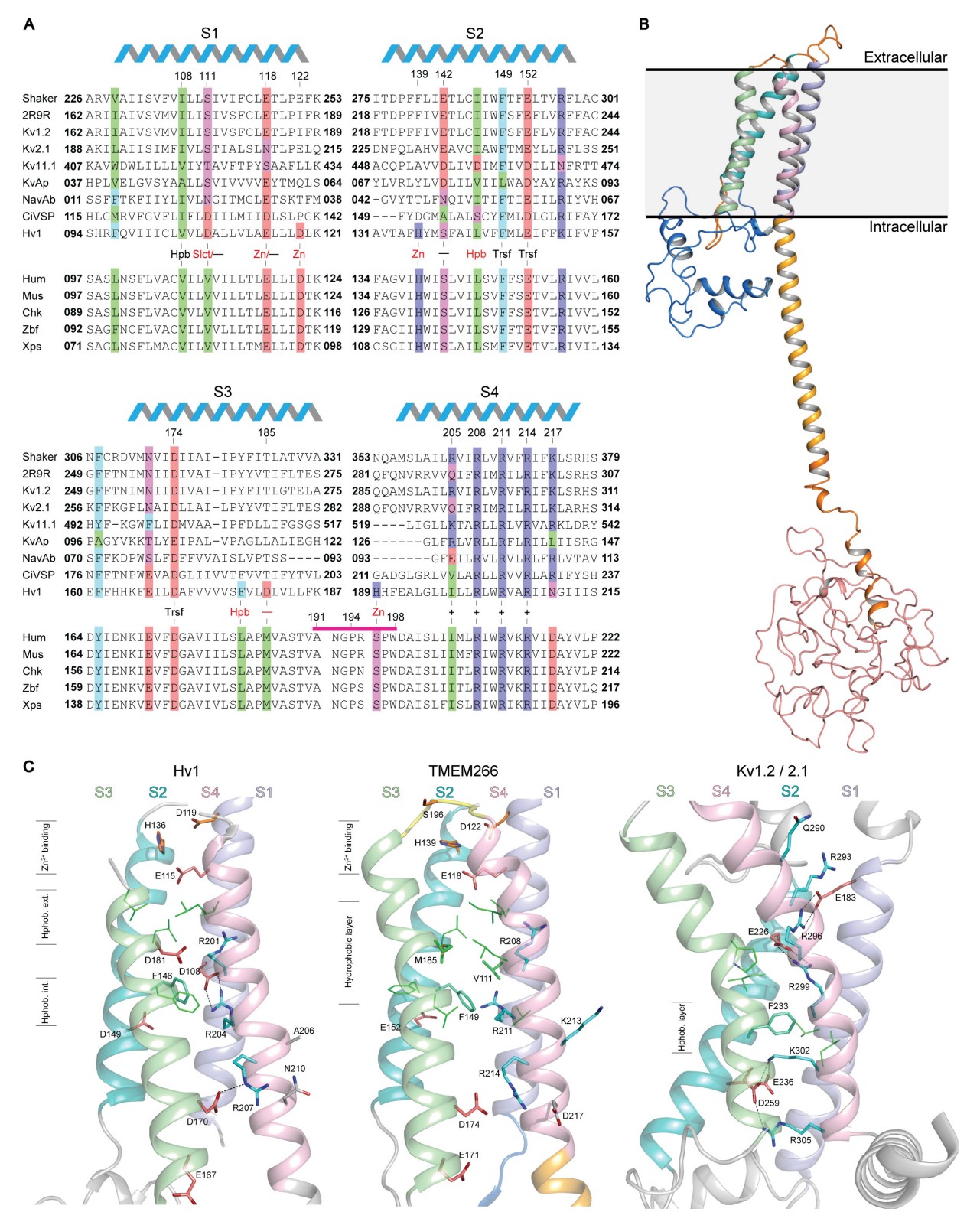

**Figure 1.** Relationship between the S1-S4 domain in hTMEM266 and those in established voltage sensors. (**A**) Multiple sequence alignment of S1-S4 voltage-sensing domains from a selection of Kv and Nav channels, Ci-VSP, Hv1 and diverse orthologs of hTMEM266. A selection of highly conserved acidic (red), basic (blue), polar uncharged (magenta), aromatic (cyan) and aliphatic (green) residues are highlighted, including the functionally important S4 Arg (+), charge-transfer center (Trsf), and acidic and/or polarizable 'counter-charges' (-) at positions facing in towards the domain core. In addition,

*Figure 1 continued on next page*

*Figure 1 continued*

selected residues for Hv1 including the proton 'selectivity filter' (Slct), zinc binding residues (Zn), and conserved acidic (-) and hydrophobic residues (Hpb) at the core of the domain are labelled in red. hTMEM266 residue numbers are marked above for reference and the helical schematic above the alignment marks the extent of the helices in the homology model of hTMEM266, while residues that were mutated to Cys for fluorimetry experiments are indicated by a magenta bar. Sequences included in the alignment (identified here by their Refseq or PDB accession #) are: Drosophila Shaker (NP_001162788.1), Kv1.2/2.1 paddle chimera (PDB ID: 2R9R), Rat Kv1.2 (NP_037102.1), Rat Kv2.1 (NP_037318.1), Human Kv11.1 (NP_000229.1), Aeropyrum pernix Kv (Genbank: BAA79939.1), Arcobacter butzleri Nav (WP_012147720.1), Ciona intestinalis VSP (NP_001028998.1.), Mouse Hv1 (NP_001035954.1), along with TMEM266 orthologs from human (NP_689548.2), mouse (NP_766511.1), chicken (XP_001233623.3), zebrafish (NP_001074141.1) and Xenopus frog (XP_018108344.1). (B) Homology model of the full-length human hTMEM266 based on Hv1 structure (PDB ID: 3WKV). Predicted domains are color coded as follows: N-terminus (blue), S1 (light blue), S2 (cyan), S3 (light green), S4 (light pink), coiled coil (orange), disordered C-terminus (red). Membrane boundaries are estimated from superposition with the Kv1.2/2.1 paddle chimera structure where detergents and lipids are resolved. (C) Structural comparison of the S1-S4 domains of the human hTMEM266 homology model with those of Hv1 (PDB ID: 3WKV) and Kv1.2/2.1 paddle chimera (PDB ID: 2R9R). Backbone coloring scheme from (B) has been preserved and the functionally important S4 Arg residues (blue), S2 phenylalanine from the charge transfer center (cyan), and stabilizing acidic counter-charges (red) are rendered as sticks with dashed lines between the side chains denoting predicted H-bonds. In the Hv1 structure, aliphatic residues (green) directed into the 4TM core form distinct hydrophobic layers above (extracellular) and below (intracellular) the charge-transfer center. In contrast, in the hTMEM266 homology model there is a single, thicker hydrophobic layer extending upwards from the level of the charge transfer center, while the hydrophobic layer is lower and thinner in the Kv1.2/2.1 structure. Finally, residues believed to contribute to the extracellular $Zn^{2+}$ binding sites are marked in orange on the Hv1 and hTEM266 structures.

DOI: https://doi.org/10.7554/eLife.42372.002

The following figure supplements are available for figure 1:

**Figure supplement 1.** Phylogenetic analysis of TMEM266.
DOI: https://doi.org/10.7554/eLife.42372.003
**Figure supplement 2.** Sequence alignment and conservation of TMEM266.
DOI: https://doi.org/10.7554/eLife.42372.004

the S2 helix and both E171 and D174 on the S3 helix) the charge transfer center that could potentially stabilize the positively charged S4 Arg residues within the membrane (*Long et al., 2007*; *Papazian et al., 1995*). Finally, hTMEM266 appears to share a $Zn^{2+}$ binding site with Hv1. In Hv1, four residues (E115 and D119 in S1, H136 in S2 and H189 in S4; numbering for mHv1) have been implicated in coordinating $Zn^{2+}$, a metal ion that inhibits the proton channel by stabilizing a closed state of the channel (*Ramsey et al., 2006*; *Takeshita et al., 2014*). Three of these four $Zn^{2+}$-coordinating residues are conserved in hTMEM266 (E118 and D122 in S1 and H139 in S2), while a Ser (S196 in S4) is present at the equivalent position of H189 in Hv1 (*Figure 1A*). Taken together, examination of the sequence of the S1-S4 domain in hTMEM266 reveals that it contains many of the most critical residues found within the S1-S4 voltage-sensing domains in other proteins, raising the possibility that it may function as a voltage sensor.

The S1-S4 sequence of hTMEM266 also has several features that appear unique. Although the 'X' amino acids in the RXXRXXR motif in the S4 helix are generally non-polar, hTMEM266 contains an additional Lys residue (K213) just before the third Arg residue (*Figure 1A,C*). In our homology model, K213 projects out from the domain (*Figure 1C*), raising the possibility that it may be involved in oligomerization (see below). The S4 helix of hTMEM266 also contains an acidic residue (D217) following and in register with the S4 Arg residues, a feature also seen in the S4 helices of EAG and ERG Kv channels (*Wang and MacKinnon, 2017*; *Whicher and MacKinnon, 2016*), but that is otherwise uncommon in voltage-sensing domains. Finally, hTMEM266 has non-polar residues (V111 in S1 and M185 in S3) in the core of the 4TM bundle that are typically charged or polar in other voltage-sensing domains, leading to a larger cluster of hydrophobic residues external to the charge transfer center (*Figure 1A,C*).

## The S4 helix of hTMEM266 can support voltage-dependent gating in Hv1 and Kv channels

Because the Hv1 proton channel is the closest relative of TMEM266, we began by confirming that hTMEM266 does not form functional proton channels when expressed in HEK cells and studied using whole-cell patch clamp recordings with solutions optimized for recording proton currents. We could readily measure robust proton currents in response to membrane depolarization for cells expressing hHv1, but observed only small leak currents for cells expressing hTMEM266 (*Figure 2A,B*) even

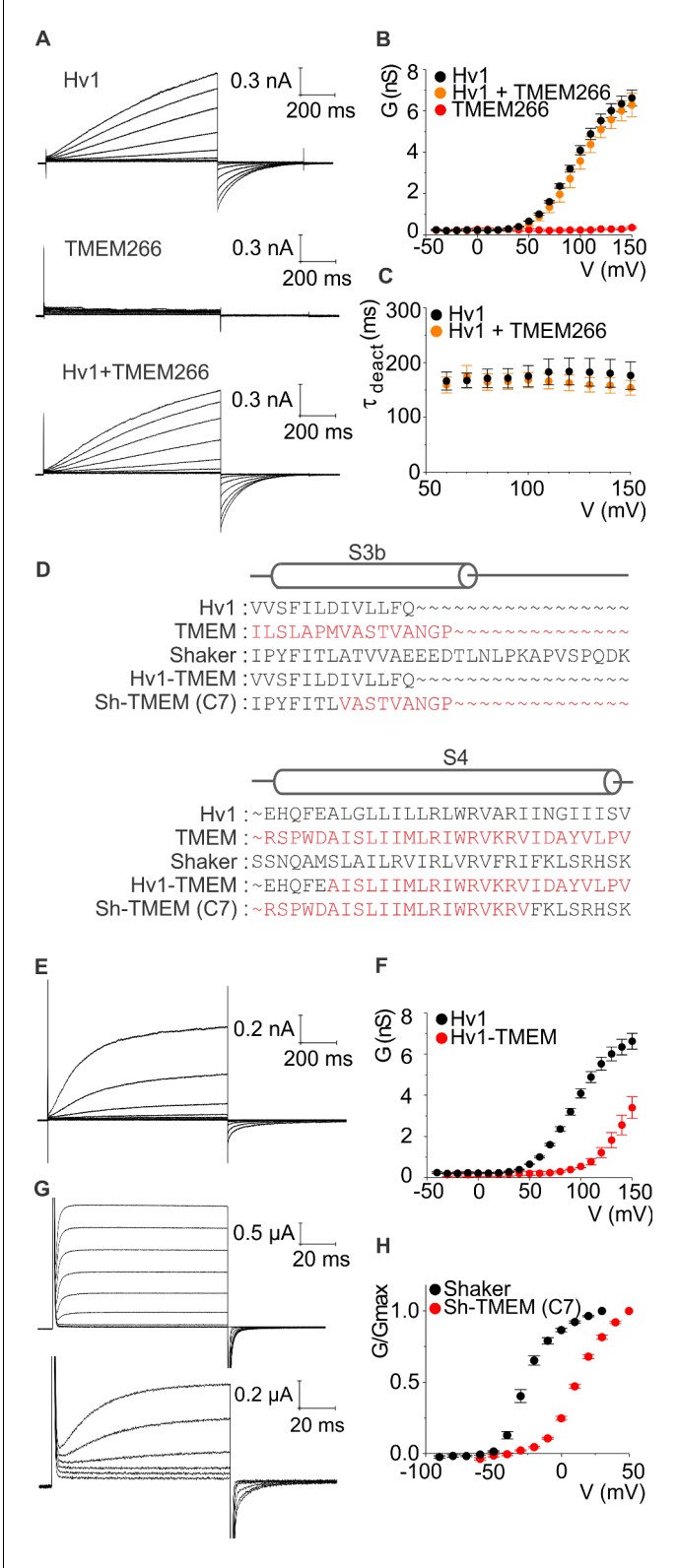

**Figure 2.** hTMEM266 does not function as a proton channel but its S4 helix can support voltage sensing in Hv1 and Kv channels. (**A**) Whole-cell macroscopic currents recorded from HEK cells expressing hHv1, hTMEM266, and hHv1 plus hTMEM266 (top to bottom) elicited using 1 s test depolarization from −40 to +150 mV in increments of 10 mV. Holding voltage was −80 mV and tail voltage was −60 mV. Currents displayed are from −30 to 150 mV in

*Figure 2 continued*

increments of 20 mV for clarity. No leak correction has been applied. (**B**) Average conductance (G)-voltage (V) relationships for hHv1 (n = 4), hTMEM266 (n = 4), and hHv1 and hTMEM266 co-transfected (n = 3). (**C**) Average deactivation time constants (τ) derived from single-exponential fits to tail currents elicited by repolarization to −60 mV from cells in B. (**D**) Sequence alignments of hHv1, hTMEM266, Shaker and two chimeras. (**E**) Proton currents recorded from a HEK cell expressing the hHv1-hTMEM266 chimera elicited using 1 s test depolarization from −40 to +150 mV in increments of 10 mV. Holding voltage was −80 mV and tail voltage was −40 mV. Currents shown are from −30 to 150 mV in increments of 20 mV for clarity. No leak correction applied. (**F**) Average G-V relationships for Hv1 (n = 4) and the hHv1-hTMEM266 chimera (n = 3). (**G**) Macroscopic K$^+$ currents recorded from oocytes expressing Shaker (top) and the Shaker-hTMEM266 chimera (bottom) elicited using 100 ms test depolarization from −90 to +80 mV in increments of 10 mV. Holding voltage was −90 mV and tail voltage was −60 mV for Shaker and −50 mV for the chimera. Current traces from −60 to 30 mV for Shaker and the Shaker-hTMEM266 chimera from −50 to 50 mV, in increments of 20 mV, are shown. (**H**) Average normalized G-V relationships for Shaker (n = 4) and Sh-hTMEM266 (n = 6). In all panels error bars represent SEM.

DOI: https://doi.org/10.7554/eLife.42372.005

The following figure supplements are available for figure 2:

**Figure supplement 1.** hTMEM266 efficiently traffics to the plasma membrane of HEK 293 cells.
DOI: https://doi.org/10.7554/eLife.42372.006
**Figure supplement 2.** Oligomerization of hTMEM266.
DOI: https://doi.org/10.7554/eLife.42372.007

though a C-terminally GFP-tagged construct of hTMEM266 traffics efficiently to the plasma membrane (*Figure 2—figure supplement 1*). Since hTMEM266 is predicted to contain a coiled-coil domain that is similar to the coiled-coil that mediates dimerization in Hv1, we also tested the possibility that hTMEM266 might be able to oligomerize with Hv1. Disruption of dimerization of Hv1 alters the gating properties of the channel, producing a pronounced speeding of channel deactivation (*Koch et al., 2008*; *Tombola et al., 2008*). We therefore tested whether hTMEM266 might be able to oligomerize with Hv1 and alter its gating properties. However, when hHv1 and hTMEM266 were expressed together, we measured voltage-activated proton currents that were indistinguishable from Hv1 alone (*Figure 2A–C*). Taken together, these results suggest that hTMEM266 is not a proton channel and does not alter the functional properties of Hv1 when the two proteins are co-expressed.

We next tested whether the S4 helix of hTMEM266 is capable of sensing membrane voltage in the context of its closest relative Hv1, and in that of the Shaker Kv channel, a particularly well-studied Kv channel that can tolerate transplantation of S4 helices from other voltage-activated ion channels (*Alabi et al., 2007*; *Bosmans et al., 2008*) (*Figure 2D*). Transfer of the S4 helix from hTMEM266 into hHv1 results in functional voltage-activated proton channels when expressed in HEK cells, with a voltage-activation relation that is shifted to more depolarized voltages compared to Hv1 (*Figure 2E,F*). We also succeeded in generating a functional chimera wherein the S3b-S4 paddle motif was transplanted from hTMEM266 into the Shaker Kv channel, and when expressed in oocytes this chimera also displayed a similarly shifted voltage-activation relation (*Figure 2G,H*). These perturbations in the energetics of voltage-dependent gating are relatively modest when considering how many residues differ between the transferred regions of hTMEM266 and either hHv1 or Shaker (*Figure 2D*) and indicate that the S4 helix of hTMEM266 can support voltage-dependent gating in the context of two distinct voltage-activated ion channels.

## hTMEM266 can form oligomers

Functional and biochemical experiments have established that the Hv1 channel forms dimers and that the C-terminal coiled-coil extension of the S4 helix mediates dimerization (*Gonzalez et al., 2010*; *Koch et al., 2008*; *Lee et al., 2008*; *Tombola et al., 2008*; *Tombola et al., 2010*). Because hTMEM266 is also predicted to contain a C-terminal coiled-coil extension of S4 (*Figure 1B*), we tested whether the protein forms oligomers when expressed in HEK 293 cells. Exposure of cells expressing hTMEM266 to disuccinimidyl substrate (DSS), a bifunctional membrane permeable primary amine crosslinking reagent, resulted in formation of a higher molecular weight band in Western blots of PAGE gels (*Figure 2—figure supplement 2A,C*), consistent with dimer formation. Because

the C-terminal coiled-coil of hTMEM266 contains a Cys residue (C252) that is conserved in Hv1 where it can form a disulfide bond in the coiled-coil (*Fujiwara et al., 2013*), we expressed the coiled-coil of hTMEM266 alone (residues 224–289) and tested whether the purified domain could form dimers. When run on a PAGE gel, the coiled-coil of hTMEM266 migrated as a single molecular species if reduced with 2-mercaptoethanol (2-ME) prior to loading (*Figure 2—figure supplement 2D*). In contrast, under non-reducing conditions a substantial fraction of the protein migrated at a higher molecular weight (*Figure 2—figure supplement 2D*), suggesting that the coiled-coil can form disulfide-mediated dimers. We also observed the presence of a higher molecular weight band in Western blots of hTMEM266 under non-reducing conditions (*Figure 2—figure supplement 2B, C*), suggesting that disulfide-mediated dimers of hTMEM266 can form. Finally, we also co-expressed N-terminally GFP-tagged and C-terminally Strep-tagged constructs of hTMEM266 and observed that the GFP-tagged construct could be pulled down along with the Strep-tagged construct when it was purified using Streptactin beads (*Figure 2—figure supplement 2E,F*). In this context, deletion of the coiled-coil containing C-terminus of hTMEM266 from the GFP-tagged construct prevented pulldown along with the Strep-tagged protein, indicating that the C-terminus of hTMEM266 mediates dimerization. From these results, we conclude that coiled-coil of hTMEM266 can mediate dimerization of the full-length protein.

## Fluorescence responses of hTMEM266 labeled at the outer end of S4

To explore whether hTMEM266 can change conformation in response to changes in membrane voltage, we introduced Cys residues at 15 positions spanning between the outer ends of the S3 and S4 helices, expressed the proteins in *Xenopus* oocytes, labeled them with (2-((5 (6)-tetramethylrhodamine)carboxylamino)ethyl methanethiosulfonate (TAMRA-MTS), and looked for voltage-dependent changes in fluorescence, an approach that has been widely used to study conformational dynamics of voltage sensors in voltage-activated ion channels (*Chanda et al., 2004*; *Chanda et al., 2005*; *Chanda and Bezanilla, 2002*; *Mannuzzu et al., 1996*; *Pathak et al., 2005*), VSPs (*Kohout et al., 2010*; *Kohout et al., 2008*; *Villalba-Galea et al., 2008*) and Hv1 (*Gonzalez et al., 2010*; *Gonzalez et al., 2013*; *Qiu et al., 2013*). Eight of the 15 positions appeared to be accessible to the extracellular solution as the fluorescence intensity after TAMRA-MTS labelling was greater than for oocytes that expressed WT hTMEM266 or were not injected with cRNA (uninjected) (*Figure 3A*). In addition, whereas the fluorescence from control oocytes (uninjected or WT hTMEM266) did not change appreciably when membrane voltage was stepped between −200 and +200 mV, for these eight positions we observed rapid fluorescence increases (dequenching) following membrane depolarization (*Figure 3B*). The size of the fluorescence change ($\Delta F/F$) ranged from 1% to 5% (*Figure 3B, C*) with nearly linear F-V relations that did not saturate over a voltage range of −200 to +200 mV (*Figure 3C,D*). Two of the positions also showed a second, slower phase in the fluorescence signals; for hTMEM266 P194C, we typically observed a rapid increase in fluorescence with depolarization, followed by a slower decrease (quenching), and for hTMEM266 W198C, we observed both rapid and slow phases of dequenching (*Figure 3C*).

To investigate the kinetics of the fast dequenching components, we measured fluorescence responses for voltage steps from −100 mV to +200 mV for several labeled positions using different modes of the oocyte clamp. For hTMEM266 G193C, a position that lacks the slow quenching component, $\tau_F$ was 408 µs in two-electrode voltage clamp mode, which is close to the value of the clamp speed estimated from the capacitive transient ($\tau_Q$=571 µs) (*Figure 3—figure supplement 1E*). Although this value of $\tau_Q$ is slower than the $\tau_V$ measured with the voltage-measuring electrode ($\tau_V$=108 µs), that electrode only provides the clamp speed at the tip of the electrode and is an overestimate of the actual clamp speed for the whole cell. In the faster cut-open voltage clamp mode, we measured a $\tau_F$ of 130 µs for the G193 position when $\tau_V$ was 39 µs. The kinetics of the fast dequenching component was also as rapid as the estimated speed of the clamp for P194C, although the presence of the slow quenching component in this case overlapped with the fast dequenching component (*Figure 3—figure supplement 1D*). Taken together, these observations lead us to conclude that the kinetics of the dequenching components for the labeled positions we studied are remarkably rapid and are likely limited by the speed of our voltage clamp.

We also examined these current recordings from cells expressing hTMEM266 G193C for any signs of charge movement due to changes in the conformation of the protein. Although we could not unambiguously detect non-linear charge movement similar to what has been measured for

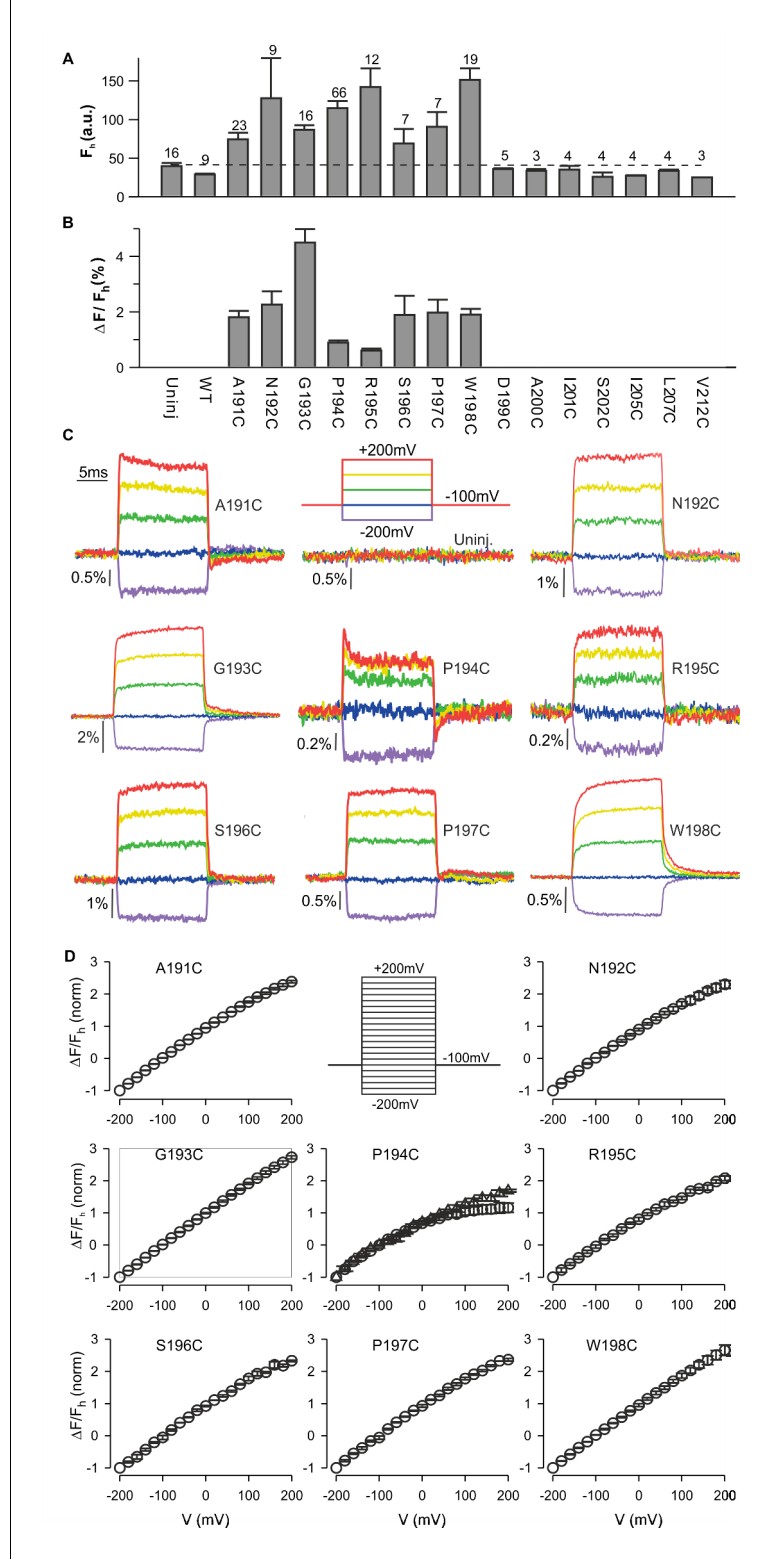

**Figure 3.** Fluorescence responses from hTMEM266 labeled at the external ends of the S3 and S4 helices. (A) Average fluorescence intensities ($F_h$, arbitrary units) of TAMRA-MTS labeled oocytes expressing hTMEM266 without (WT) or with Cys substitutions at different positions between the external ends of S3 and S4. Holding voltage was −100 mV. Dashed line represents the average fluorescence intensity recorded from uninjected oocytes (Uninj). Number of oocytes is indicated above each bar. (B) Average change in fluorescence intensity (Δ*F*/

*Figure 3 continued on next page*

*Figure 3 continued*

$F_h$) for all positions studied. $\Delta F$ is the change in fluorescence intensity for a voltage step from $-100$ mV to $+200$ mV and $F_h$ is the fluorescence intensity at $-100$ mV. The number of oocytes is indicated above the corresponding bar in A. (**C**) Representative fluorescence signals recorded from uninjected oocytes and oocytes expressing Cys mutants of hTMEM266 after labeling with TAMRA-MTS. Voltage protocol with color code is shown in the top middle panel. (**D**) Normalized average fluorescence-voltage ($\Delta F/F_h$-V) relations for labeled constructs ($n = 3$ oocytes). $\Delta F/F_h$ was normalized to the value measured at $-200$ mV. Circles represent $\Delta F_e/F$ where $\Delta F_e$ is the difference between the average fluorescence at the end (last 5 ms) of the depolarization step and the baseline fluorescence at $-100$ mV. For the P194C panel triangles represent $\Delta F_i/F$ where $\Delta F_i$ is the difference between the baseline fluorescence at $-100$ mV and the initial fluorescence at the beginning of the depolarization step. In all panels error bars represent SEM.

DOI: https://doi.org/10.7554/eLife.42372.008

The following figure supplements are available for figure 3:

**Figure supplement 1.** Fluorescence responses for Shaker, Ci-VSP, Ci-Hv1 and hTMEM266.
DOI: https://doi.org/10.7554/eLife.42372.009

**Figure supplement 2.** Transient currents from oocytes expressing hTMEM266.
DOI: https://doi.org/10.7554/eLife.42372.010

voltage-sensor activation in voltage-activated ion channels (*Bezanilla, 2018*) or VSPs (*Murata et al., 2005*), the amount of charge measured by integrating transient capacitive currents was larger in oocytes expressing hTMEM266 G193C compared to control un-injected oocytes (*Figure 3*; *Figure 3—figure supplement 2*). This apparent increase in membrane capacitance suggests either that hTMEM266 contains mobile charges, or that expression of hTMEM266 changes the properties of the plasma membrane, such as its area, thickness or polarizability.

## Probing for conformational changes in hTMEM266 with fluorescence quenchers

The rapid kinetics and near linear F-V relationships observed for most sites in hTMEM266 are quite distinct from TAMRA-MTS labeled Cys mutants of the Shaker Kv channel, Ci-VSP and Hv1 (*Figure 3—figure supplement 1A–C*), and raises the possibility that the fluorophore in our hTMEM266 labeled constructs is positioned within the membrane electric field, which might directly alter fluorescence by perturbing the ground or excited energy states of the fluorophore, a mechanism that has been termed the Stark-shift or electrochromic effect (*Asamoah et al., 2003*; *Klymchenko and Demchenko, 2002*; *Klymchenko et al., 2006*).

To further investigate whether hTMEM266 undergoes a rapid conformational rearrangement in response to changes in membrane voltage, we examined the effects of collisional quenchers to see if we could detect voltage-dependent changes in accessibility of quenchers to TAMRA (*Cha and Bezanilla, 1998*; *Chanda et al., 2004*). We first tested a variety of candidate ions and molecules for quenching of TAMRA-MTS in solution and found that anionic I$^-$ and zwitterionic tryptophan (Trp) were effective quenchers at low mM concentrations, and that Ni$^{2+}$ was effective at higher concentrations (*Figure 4A,B*). In control experiments with TAMRA-MTS labeling of uninjected oocytes, fluorescence quenching from extracellular Trp or I$^-$ was independent of membrane voltage (*Figure 4C*; *Figure 4—figure supplement 2*). In contrast, for cells expressing G193C labeled with TAMRA-MTS, extracellular application of Trp quenched the fluorescence more efficiently when the membrane was depolarized, consistent with the fluorophores becoming more accessible to the quencher as the membrane voltage was increased (*Figure 4D*). For each construct, the fraction of fluorescence quenched ($f_q$) was characterized at the holding voltage of $-100$ mV, $f_q$ ($-100$ mV)$=1 - F_q$ ($-100$ mV)/$F_c$ ($-100$ mV), where $F_q$ and $F_c$ are the fluorescence intensities measured in the presence and absence of the quencher, respectively. The voltage-dependence of quenching was characterized by $\Delta f_q$ ($+200$ mV)/$f_q$ ($-100$ mV), the change in quenching upon depolarization to $+200$ mV. Comparison of the results of Trp quenching experiments for all eight positions studied reveals a comparable amount of quenching for all sites, but site-specific variations in the voltage-dependence of quenching with the largest changes for G193C, S196C and P197C, intermediate changes for N192C, P194C and R195C, and a markedly reduced voltage-dependence for A191C and W198C (*Figure 4E,F*; *Figure 4—figure supplement 1*). We also undertook equivalent experiments with extracellular

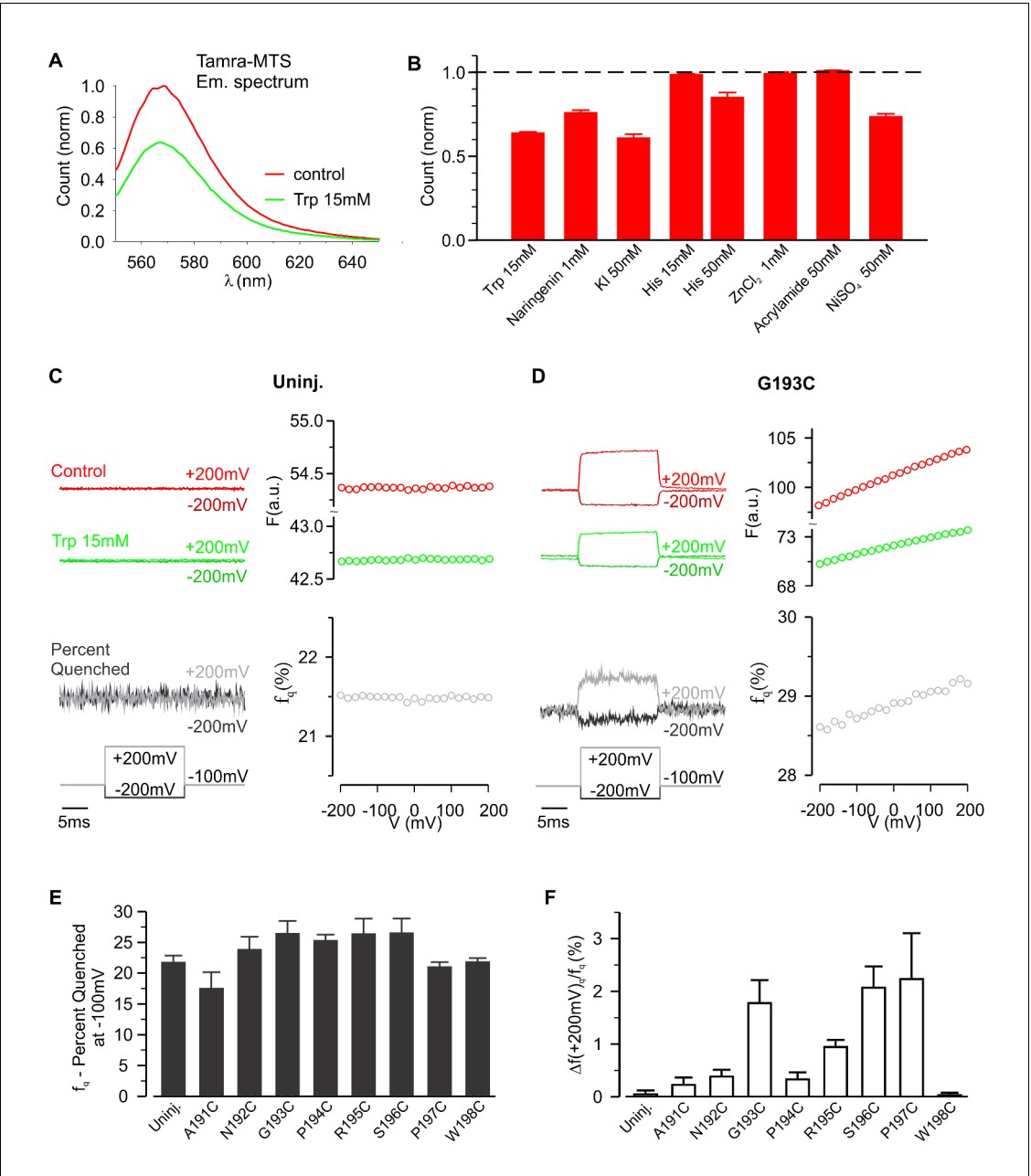

**Figure 4.** Voltage-dependent tryptophan quenching of TAMRA-labeled hTMEM266 constructs. (**A**) Quenching of TAMRA-MTS fluorescence by tryptophan (15 mM) in aqueous solution. (**B**) Normalized average solution TAMRA-MTS fluorescence in the presence of potential quenchers or modulatory ions. (**C**) Effect of 15 mM tryptophan on fluorescence from an uninjected oocyte labeled with TAMRA-MTS. Examples of voltages steps from −100 mV to −200 mV or +200 mV are shown on the left, with the fluorescence intensity in the absence (red) and presence (green) of 15 mM tryptophan plotted above the resulting percent of fluorescence quenched ($f_q$, grey). The voltage dependence of the fluorescence and fluorescence quenching are plotted on the right. (**D**) Effect of 15 mM tryptophan on fluorescence from an hTMEM266 G193C expressing oocyte labeled with TAMRA-MTS. (**E**) Average percent of fluorescence quenched by 15 mM tryptophan, $f_q$, at the holding voltage (−100 mV) for TAMRA-MTS labeled hTMEM266 constructs with Cys substituted at the external ends of S3 and S4, as well as for uninjected oocytes (n = 4 to 7 oocytes). (**F**) Average voltage-dependent change in quenching at +200 mV, $\Delta f_q(+200 \text{ mV})/f_q(−100 \text{ mV})$, for TAMRA-MTS labeled hTMEM266 constructs with Cys substituted at the external ends of S3 and S4, as well as for uninjected oocytes (n = 4 to 7 oocytes). See *Figure 4—figure supplement 1* for primary data for positions other than G193C. In all panels error bars represent SEM.

DOI: https://doi.org/10.7554/eLife.42372.011

*Figure 4 continued on next page*

*Figure 4 continued*

The following figure supplements are available for figure 4:

**Figure supplement 1.** Voltage-dependent tryptophan quenching of TAMRA-labeled Cys- substituted hTMEM266 for different positions at the external sides of the S3 and S4 helices.
DOI: https://doi.org/10.7554/eLife.42372.012

**Figure supplement 2.** Iodide quenching of TAMRA-labeled hTMEM266.
DOI: https://doi.org/10.7554/eLife.42372.013

**Figure supplement 3.** Iodide quenching of TAMRA attached to hTMEM266 at different positions on the external ends of the S3 and S4 helices.
DOI: https://doi.org/10.7554/eLife.42372.014

application of I$^-$ and observed comparable results; quenching was enhanced by membrane depolarization and the voltage dependence of quenching was site-specific (*Figure 4—figure supplements 2* and *3*). However, quenching by the negatively charged I$^-$ was more voltage-dependent than quenching by the neutral Trp (compare *Figure 4F* for Trp with *Figure 4—figure supplement 2E* for I$^-$), suggesting that the electric potential near the fluorophore differs from the bulk solution. Collectively, these experiments with collisional quenchers strongly suggest that hTMEM266 undergoes a rapid conformational change with membrane depolarization.

## Slow conformational changes detected with labeling at P194C

The ability of a fluorophore to detect protein conformational changes depends on its point of attachment. As described above, for two of the eight attachment positions (P194C and W198C) depolarization caused both a rapid initial change in fluorescence followed by a slower secondary response, suggesting that the protein can undergo two distinct conformational changes (*Figure 3C*). These two components can be most easily distinguished for P194C, where the initial rapid fluorescence increase (i.e. dequenching) was followed by a second, slower decrease (i.e. quenching) (*Figure 5A*). Furthermore, after returning to the holding potential at the end of the depolarizing pulse, the fluorescence overshot (i.e. 'tail fluorescence') before rising back to its initial value as the protein returned to its resting conformation. As shown in *Figure 5A*, the amplitude of the slower component, $\Delta F_{slow}$, can be characterized by the difference between fluorescence levels at the beginning ($\Delta F_i$) and end ($\Delta F_e$) of the voltage step. Curiously, the amplitude and time constant of the slower component varied between cells (see *Figure 6B–D*). However, for most cells the slow component could be easily distinguished and showed a non-linear dependence on voltage. Whereas the rapid component was roughly linear from −200 mV to +200 mV (*Figure 5B*), the slow component was essentially undetectable during hyperpolarizing steps, but increased rapidly when the membrane was depolarized without any apparent saturation (*Figure 5C*).

The slower conformational change also modulated quenching of TAMRA attached to P194C by extracellular Trp and I$^-$ (*Figure 5D–I*). Fluorescence quenching rapidly increased at the start of each depolarization step, but then decreased during the step suggesting that the slow conformational change reduced the accessibility of the fluorophore to the quencher (*Figure 5D–I*). This hypothesis is also consistent with the 'tail fluorescence' of the example shown in *Figure 5G*, in which the fluorescence quenching fraction is at a minimum immediately after repolarizing to −100 mV. Collectively, the results presented thus far are consistent with hTMEM266 undergoing two conformational rearrangements in response to changes in membrane voltage. The first occurs rapidly and results in an enhanced accessibility of the fluorophore to collisional quenchers. The second occurs on the time-scale of milliseconds and reduces TAMRA fluorescence attached at P194C. This second quenching component is not observed below 0 mV but becomes increasingly apparent as the voltage becomes more depolarized.

## Modulation of slow fluorescence responses by Zn$^{2+}$

The Hv1 voltage-activated proton channel is inhibited by micromolar concentrations of Zn$^{2+}$ and mutagenesis experiments have implicated two His residues (H136 near S2 and H189 near S4) in forming the Zn$^{2+}$ binding site (*Cherny and DeCoursey, 1999*; *Musset et al., 2008*; *Ramsey et al., 2006*; *Sasaki et al., 2006*). The Hv1 crystal structure is compatible with these His residues contributing to the Zn$^{2+}$ binding site, but also reveals the likely involvement of two acidic residues (E115 and

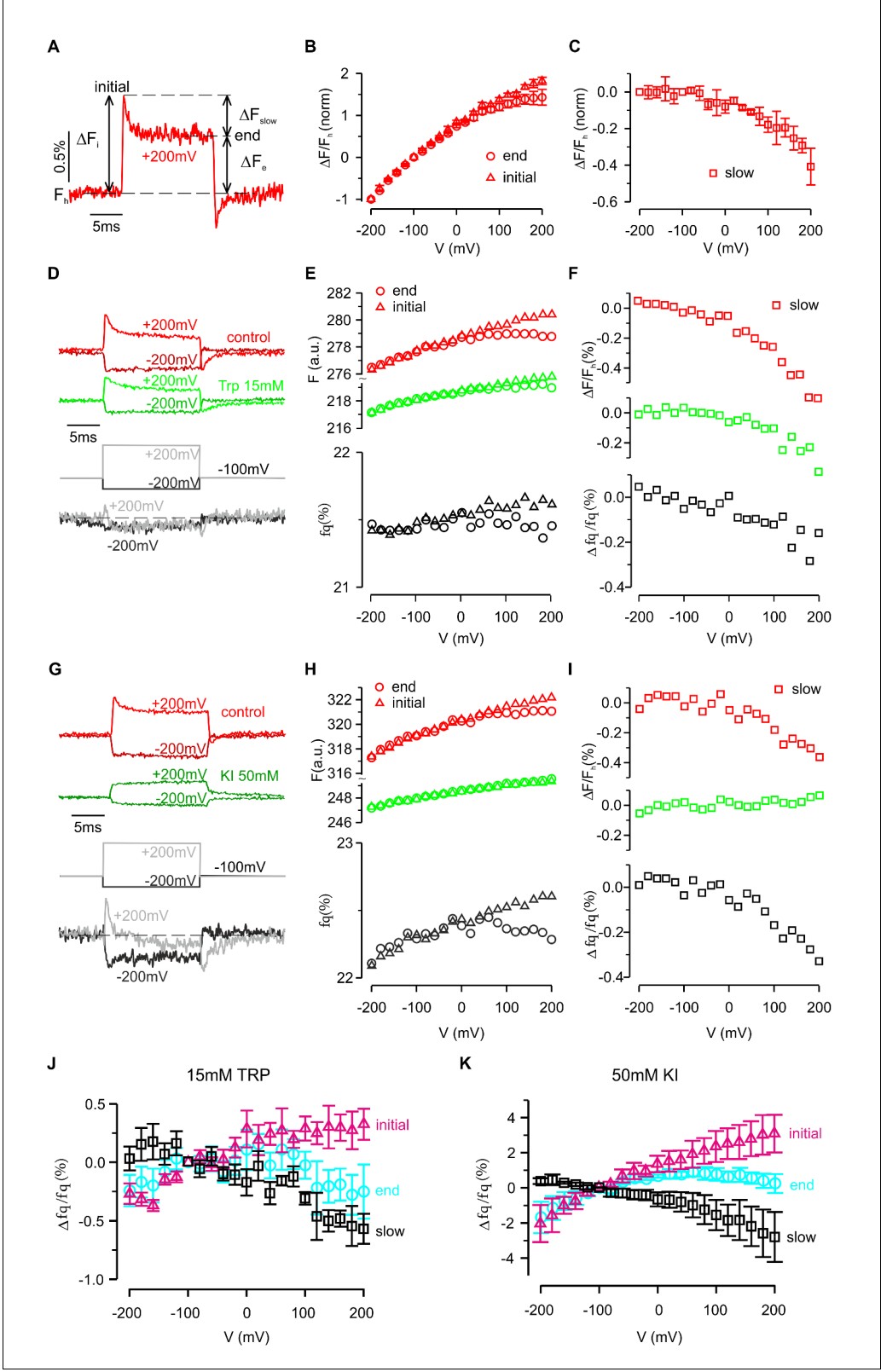

**Figure 5.** Fluorescence and quenching of hTMEM266 labeled with TAMRA-MTS at P194C. (**A**) Representative fluorescence response to a voltage step from −100 mV to +200 mV for a hTMEM266 P194C expressing oocyte labeled with TAMRA-MTS. $F_h$ is the fluorescence at the holding voltage of −100 mV, $\Delta F_i$ is the initial change in fluorescence at the beginning of the voltage step, $\Delta F_e$ is change in fluorescence at the end of the voltage step,

*Figure 5 continued on next page*

*Figure 5 continued*

and $\Delta F_{slow}$ is the change in fluorescence during the voltage pulse ($\Delta F_{slow} = \Delta F_i - \Delta F_e$). (**B**) Average $\Delta F/F_h$ as a function of voltage normalized to the value of $\Delta F/F_h$ at −200 mV. Triangles show the initial change, $\Delta F_i/F_h$, and circles represented the change and the end of the voltage step, $\Delta F_e/F_h$ (n = 3 oocytes). (**C**) Average slow fluorescence intensity change during the voltage step, $\Delta F_{slow}/F_h$, as a function of membrane voltage (n = 3 oocytes). (**D–F**) Voltage dependent fluorescence quenching by 15 mM tryptophan. Examples of voltages steps from −100 mV to −200 mV or +200 mV are shown in panel D, with the fluorescence intensity in the absence (red) and presence (green) of 15 mM tryptophan plotted above the resulting percent of fluorescence quenched ($f_q$, grey). Panel E shows the initial (triangles) and final (circle) fluorescence intensity during each voltage step in control (red) and 15 mM Trp (green) along with the corresponding percent of fluorescence quenched by tryptophan (grey). Finally, the magnitude of the slow component of the fluorescence ($\Delta F_{slow}/F_h$) in control (red) and 15 mM Trp (green), along with the resulting quenched fraction ($\Delta f_q/f_q$) is shown in F. (**G–I**) Voltage dependent quenching by 50 mM KI. (**J**) The average initial (triangles), end (circles) and slow change (squares) in fluorescence quenching by 15 mM tryptophan as a function of voltage (*n* = 7 oocytes). (**K**) The average initial (triangles), end (circles) and slow change (squares) in fluorescence quenching by 50 mM Iodide as a function of voltage (*n* = 5 oocytes). In all panels error bars represent SEM.

DOI: https://doi.org/10.7554/eLife.42372.015

D119) near the external end of S1 (*Takeshita et al., 2014*). Because three of the four residues implicated in coordinating $Zn^{2+}$ in Hv1 are conserved in hTMEM266 (E118 and D122 in S1 and H139 in S2), we investigated whether extracellular $Zn^{2+}$ could modulate the fluorescence signals from TAMRA-labeled constructs. The effects of extracellular $Zn^{2+}$ application are shown for TAMRA-labeled uninjected oocytes and for two examples of labeled cells expressing hTMEM266 P194C (*Figure 6A–C*). For uninjected oocytes, 100 μM $Zn^{2+}$ produced only very small decreases in fluorescence (*Figure 6E*) that was constant at all voltages examined. Even at millimolar concentrations, $Zn^{2+}$ has no measurable effect on TAMRA-MTS fluorescence in solution (*Figure 4B*), and therefore we think the small decreases in fluorescence represent bleaching of the fluorophore that occurred between the control recording and that after application of $Zn^{2+}$. In contrast, for P194C labeled hTMEM266-expressing cells, $Zn^{2+}$ produced a dramatic decrease in the fluorescence intensity at the holding voltage of −100 mV (*Figure 6E*). In addition, although the relative amplitude of the initial fast ($\Delta F_i$) and second slower ($\Delta F_{slow}$) voltage-dependent changes in fluorescence intensity varied between oocytes (*Figure 6D*), the addition of $Zn^{2+}$ greatly enhanced the fast fluorescence change after a depolarization while the slow component decreased (*Figure 6F*). We also explored the effects of $Zn^{2+}$ on the rapid dequenching of fluorescence observed for other labeled positions (*Figure 6—figure supplement 1*). $Zn^{2+}$ produced relatively modest decreases in fluorescence for each of the other labeled positions, and for A191C, G193C, S196C and P197C produced a modest increase in the steepness of $\Delta F/F_h$ relations (*Figure 6—figure supplement 1*). Taken together, these results suggest that $Zn^{2+}$ binds to hTMEM266 and modulates fluorescence signals in a site-specific fashion, with the effects on P194C standing out as the most robust.

To explore the mechanism by which $Zn^{2+}$ modulates fluorescence signals for TAMRA-MTS labeling of P194C, we first titrated the effects of $Zn^{2+}$ and found that its binding site in hTMEM266 has low micromolar affinity (*Figure 7A,D*), somewhat lower affinity than measured in Hv1 (*Ramsey et al., 2006*), consistent with the fact that hTMEM266 contains three of the four residues implicated in Hv1 (*Takeshita et al., 2014*). Because His is a quencher of TAMRA fluorescence (*Figure 4B*)(*Chen et al., 2010a*), we hypothesized that H139 at the outer end of TM2 in hTMEM266 may be involved in forming the $Zn^{2+}$ binding site and that the binding of $Zn^{2+}$ might modulate His quenching of TAMRA fluorescence at P194C. We mutated H139 to Ala in the P194C background and observed only small fluorescence changes with membrane depolarization for TAMRA-MTS labeled cells, consistent with that residue playing a role in both the rapid dequenching and slow quenching responses observed at P194C. In addition, extracellular application of $Zn^{2+}$ at concentrations as high as 1 mM produced only small decreases in fluorescence for that construct, without altering the slope of the $\Delta F/F$ relation (*Figure 7B,E*). We also mutated H139 to Trp since Trp is also a strong quencher (*Chen et al., 2010a*) and observed that the initial rapid fluorescence increase after depolarization was followed by a further slow increase during the voltage step (*Figure 7C*), in marked contrast to the slow fluorescence decrease observed when the native His was present at position 139. Addition of

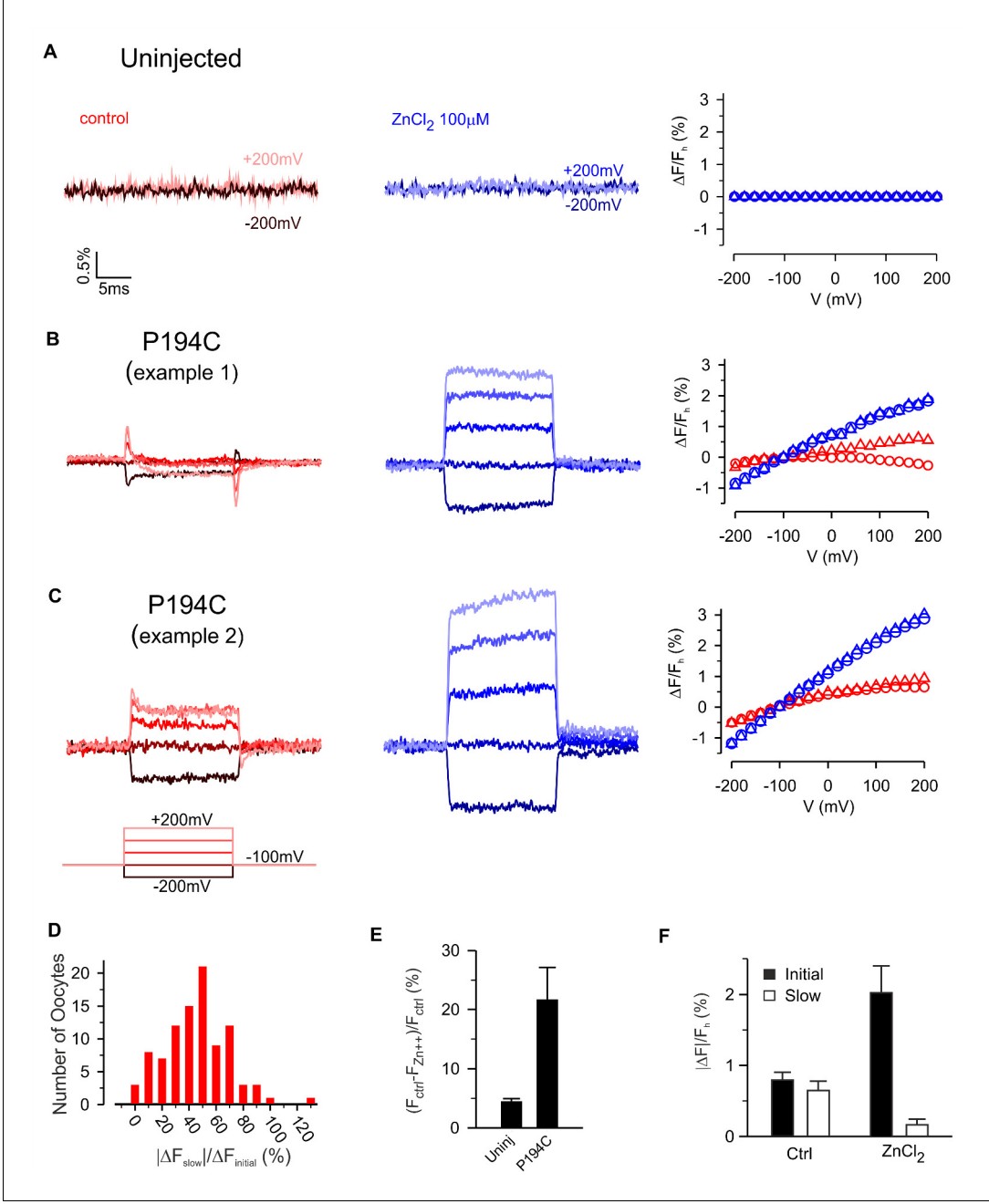

**Figure 6.** $Zn^{2+}$ modulation of fluorescence from hTMEM266 labeled with TAMRA-MTS at P194C. (**A**) Fluorescence from an uninjected oocyte labeled with TAMRA-MTS as voltage steps are first applied in control solution (red) and then in the presence of extracellular $ZnCl_2$ (blue). The voltage dependence of the fluorescence, $\Delta F/F_h$, is shown to the right. (**B,C**) Two examples of fluorescence from oocytes expressing hTMEM266 P194C labeled with TAMRA-MTS during voltage steps in control solution (red) and with $ZnCl_2$ (blue). The rapid initial change in fluorescence intensity during the first 1 ms after the voltage step (triangles) and the average value during the last 5 ms (circles) of each voltage step are plotted on the right. (**D**) Histogram illustrating how the size of the slow fluorescence change during the voltage step ($\Delta F_{slow}$) relative to the rapid initial change ($\Delta F_{initial}$) varies between oocytes. Both slow and initial components were evaluated for a step to +200 mV. (**E**) Average reduction in baseline fluorescence intensity after applying 100 μM $ZnCl_2$, ($F_{ctrl} - F_{Zn2+})/F_{ctrl}$, for uninjected oocytes (n = 3) and oocytes expressing P194C (n = 7). (**F**) Average magnitude of the initial (dark) and slow (light) fluorescence intensity changes for oocytes expressing P194C following depolarization to +200 mV in control solution and 100 μM $ZnCl_2$ (n = 5 oocytes). In all panels error bars represent SEM.

DOI: https://doi.org/10.7554/eLife.42372.016

*Figure 6 continued on next page*

*Figure 6 continued*

The following figure supplement is available for figure 6:

**Figure supplement 1.** Modulation of fluorescence responses by $Zn^{2+}$ for hTMEM266 labeled with TAMRA-MTS at different positions on the external ends of the S3 and S4 helices.

DOI: https://doi.org/10.7554/eLife.42372.017

extracellular $Zn^{2+}$ also inhibited the slow dequenching component, and the affinity of $Zn^{2+}$ appeared

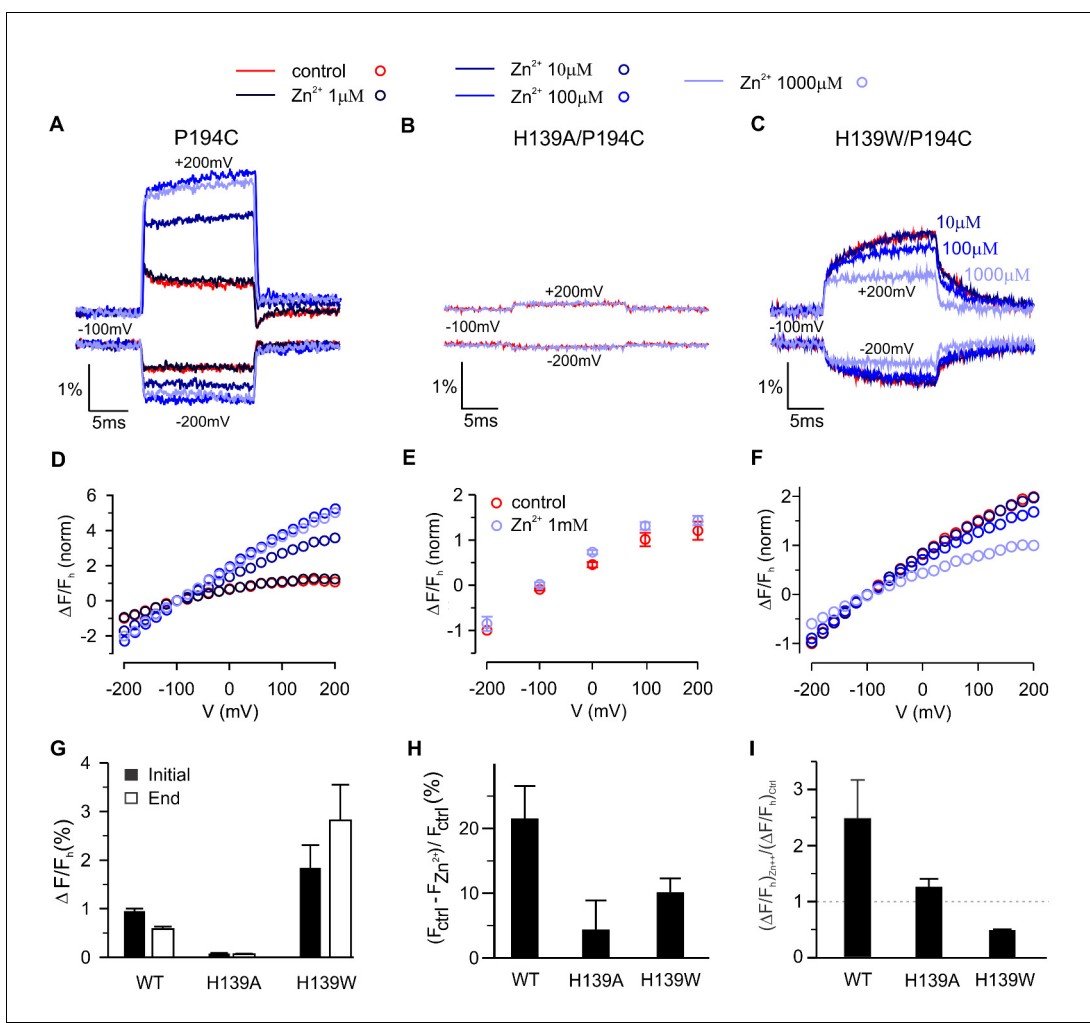

**Figure 7.** Contribution of H139 to $Zn^{2+}$ sensitivity of hTMEM266 labeled with TAMRA-MTS at P194C. (**A–C**) Fluorescence intensity during voltage steps from −100 mV to ±200 mV for oocytes expressing hTEME266 P194C (**A**), the double mutant H139A/P194C (**B**), and the double mutant H139W/P194C (**C**) for control and zinc-containing solutions. (**D–F**) Voltage dependence of the fluorescence change at the end of the voltage pulse ($\Delta F_e/F_h$) for P194C (**D**), H139A/P194C (**E**), and H139W/P194C (**F**). To permit comparison between the different constructs, the data from individual oocytes were normalized so that $\Delta F_e(-200 \text{ mV})/F_h = -1$ in control solution. (**G**) Average change in fluorescence ($\Delta F_e/F_h$), at the beginning (black) and end (white) of a step to +200 mV in control solution for P194C (n = 4 oocytes), H139A/P194C (n = 3 oocytes), and H139W/P194C (n = 3 oocytes). (**H**) Average decrease in baseline fluorescence at −100 mV, $(F_{ctrl} - F_{Zn2+})/F_{ctrl}$, upon switching from control to 1 mM $Zn^{2+}$ solution. (**I**) Effect of 1 mM $Zn^{2+}$ on the average change in fluorescence at the end of a step to +200 mV. For each construct the average change in a 1 mM $Zn^{2+}$ solution ($\Delta F_e/F)_{Zn2+}$ is normalized by the average change in control solution $(\Delta F_e/F)_{ctrl}$ (n = 4 oocytes for P194C, n = 3 oocytes for P194C/H139A, n = 3 oocytes for P194C/H139W). In all panels error bars represent SEM.

DOI: https://doi.org/10.7554/eLife.42372.018

to be lower (*Figure 7F–I*). Taken together, these results lead us to conclude that H139 contributes to forming the $Zn^{2+}$ binding site and to both the fast and slow fluorescence responses observed when P194C is labeled with TAMRA-MTS. The effects of $Zn^{2+}$ on P194C for both H139A and H139W constructs suggest that $Zn^{2+}$ binding modulates the interactions between quenching residues and the fluorophore.

## Detecting conformational changes with GFP linked to the C-terminus of S4

To interrogate voltage-dependent conformation changes in hTMEM266 using another approach, we deleted the C-terminus and introduced a variant of green fluorescent protein (super ecliptic pHluorin A227D) at the end of the S4 helix after Q233, effectively taking the same strategy used to create the VSP-based Arclight (*Jin et al., 2012*). Although expression of this hTMEM266-GFP construct produced only small voltage-dependent fluorescence changes, they qualitatively resembled those observed for the P194C construct labeled with TAMRA-MTS, showing rapid fluorescence dequenching with membrane depolarization followed by slower fluorescence quenching (*Figure 8A*). As observed with the external positions labeled with TAMRA-MTS, the fast dequenching component observed with hTMEM266-EGFP exhibited a nearly linear $\Delta F/F$-V relation (*Figure 8C*). The slow quenching component was approximately a factor of 10 slower in the hTMEM266-EGFP construct when compared to that seen in TAMRA-MTS labeled P194C construct, which may be related to the deletion of the C-terminus or the addition of the bulky EGFP protein. We also investigated the influence of extracellular $Zn^{2+}$, which produced a clearly detectable effect on the slow quenching component (*Figure 8B*), enhancing the amplitude of the slow component at voltages where it was detectable under control conditions, and appears to shift the slow component towards more negative voltages (*Figure 8*). $Zn^{2+}$ also appears to inhibit the fast fluorescence signals (*Figure 8D*), although this could be due to enhancement of the slow component that has opposite polarity. Although the fluorescence changes for this construct are small and challenging to study quantitatively, they support the conclusion that hTMEM266 undergoes both rapid and slow voltage-dependent conformational rearrangements.

## Discussion

hTMEM266 is most heavily expressed in the central nervous system, but it does not appear to function as an ion channel and its biological role has been a mystery (*Kim et al., 2014*). The goal of the present study was to determine whether the S1-S4 domain found in hTMEM266 can sense membrane voltage. We generated functional voltage-activated $H^+$ and $K^+$ channels by transplantation of the S4 helix from hTMEM266 into the Hv1 and Shaker Kv channels, respectively, demonstrating that the S4 helix of hTMEM266 is capable of sensing membrane voltage within the context of the S1-S4 domains of those channels. The fluorescence responses we observed when introducing Cys residues between the external ends of S3 and S4 and labeling with TAMRA were unusual, requiring extensive characterization to understand the underlying mechanism. For all the positions where we could label the protein, we observed increases in fluorescence with membrane depolarization that were as rapid as the speed of our voltage clamp; in the case of G193C where we used the cut-open configuration to optimize the clamp speed, the increase in fluorescence with membrane depolarization occurred with a time constant of 130 µs. In addition, for most of the positions studied we observed nearly linear $\Delta F/F$-V relationships over a ± 200 mV range of membrane voltages. The rapid kinetics and nearly linear $\Delta F/F$-V relationships raise the possibility that the fluorescence increases have an electrochromic origin (*Asamoah et al., 2003*; *Klymchenko and Demchenko, 2002*; *Klymchenko et al., 2006*). However, we are unaware of any reports of electrochromic signals recorded using TAMRA-MTS, which has no net charge, and only small electrochromic signals have been reported for the related cysteine-reactive fluorophore, tetramethylrhodamine maleimide (*Dekel et al., 2012*). Furthermore, for most TAMRA-labeled Cys mutants the fraction of fluorescence quenched by Trp, $f_q$, (*Figure 4*; *Figure 4—figure supplement 1*) increased following membrane depolarization, consistent with TMEM266 undergoing a voltage-dependent conformational change that increases the accessibility of the fluorophore to neutral quencher molecules in the extracellular solution. Importantly, the $f_q$-V relations (*Figure 4*, *Figure 4—figure supplements 1–3*) matched the nearly linear rapid fluorescence response we recorded (*Figure 3*). Interestingly, quenching by the negatively charged $I^-$ had a

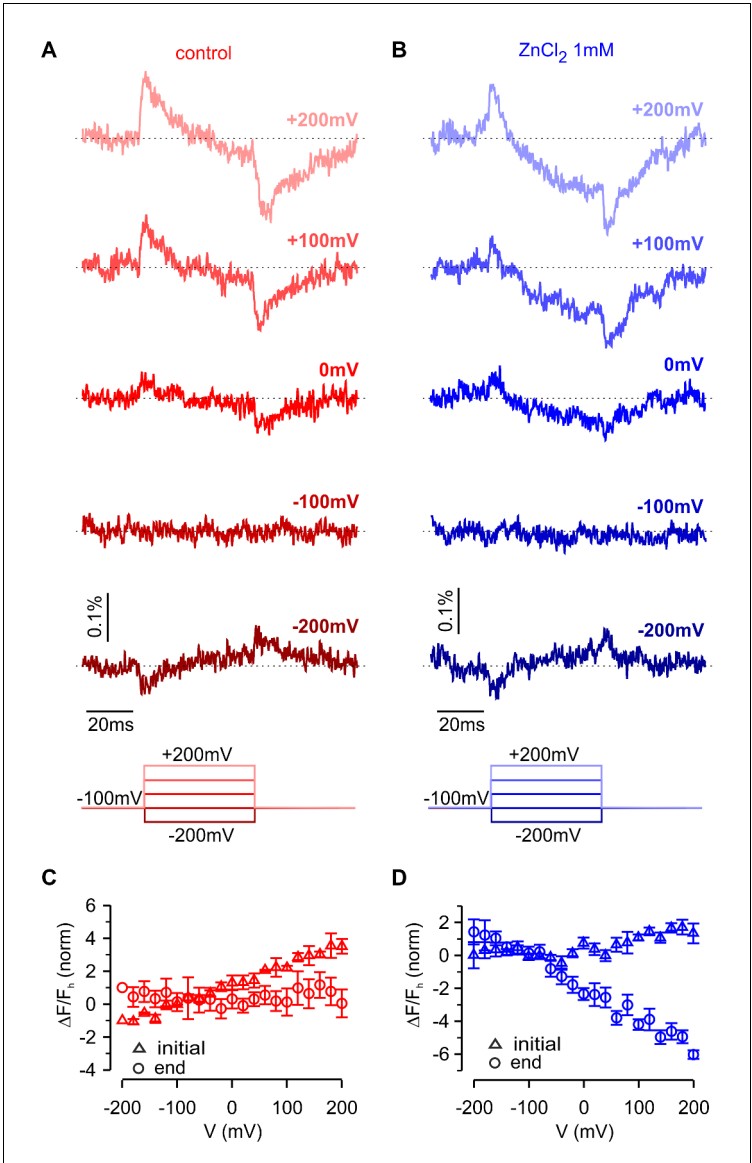

**Figure 8.** Fluorescence responses and modulation by $Zn^{2+}$ for hTMEM266 tagged with super ecliptic pHluorin GFP at the C-terminus of the S4 helix. (**A,B**) Fluorescence responses during voltage steps for an oocyte expressing hTMEM266-EGFP in control solution (red, (**A**) and after extracellular application of $ZnCl_2$ (blue, (**B**). (**C, D**) Average change in fluorescence ($\Delta F/F_h$) at the start (triangles) and end (circles) of voltage steps for the control solution (red, (**C**) and after application of $ZnCl_2$ (blue, (**D**). For each oocyte $\Delta F/F_h$ was normalized to the value measured at −200 mV in control solution ($n = 3$ oocytes). In all panels error bars represent SEM.

DOI: https://doi.org/10.7554/eLife.42372.019

steeper voltage-dependence than neutral Trp (*Figure 4*; *Figure 4—figure supplement 2*), suggesting that the electric potential near the fluorophore differs from the external solution. Finally, in our TMEM266-EGFP construct in which the fluorophore (EGFP) resides in the intracellular solution (and thus should not be subject to direct electrochromic effects), we again observed rapid changes in fluorescence and a nearly linear $\Delta F/F$-V relations (*Figure 8*), consistent with TMEM266 undergoing a rapid conformational change in response to membrane voltage. TMEM266 also appears to undergo a slower conformational change that was most evident when labeling P194C with TAMRA-MTS or when studying our hTMEM266-EGFP construct. In both cases the slow fluorescence change was only

evident for depolarizations above 0 mV but did not appear to saturate even for depolarized voltages as high as +200 mV (*Figures 4* and *8*).

Our results also demonstrate that hTMEM266 contains a regulatory $Zn^{2+}$ binding site on the external side of the S1-S4 domain that is related to the $Zn^{2+}$ binding site found in Hv1. For P194C labeled with TAMRA-MTS, external $Zn^{2+}$ ions quenched the rapid fluorescence signals at all voltages, inhibited the slow quenching component and steepened the ΔF/F relation (*Figure 6*). Although these effects of the metal ion are complex, our results show that H139 in S2 is a quenching residue that is largely responsible for the fluorescence signals measured when labeling P194C with TAMRA-MTS, and that H139 also participates in forming the $Zn^{2+}$ binding site, leading us to conclude that $Zn^{2+}$ binding modifies the interactions between H139 and TAMRA. Interestingly, mutation of H139 to Trp, another strong quencher of TAMRA fluorescence, altered the slow component of fluorescence change from quenching to dequenching in P194C labeled with TAMRA-MTS. One explanation for this observation is that the H139W mutant alters the position of the quenching residues relative to the fluorophore, effectively altering the state in which quenching is maximal. Interestingly, the effects of $Zn^{2+}$ on TAMRA-MTS labeled P194C and the hTMEM266-EGFP constructs were clearly distinct. In hTMEM266-EGFP the metal ion augmented the slow quenching component observed at positive membrane voltages and appeared to shift the component to more negative voltages (*Figure 8*). Since $Zn^{2+}$ binds on the external side of the protein and GFP was inserted into the intracellular C-terminus of the protein, these results indicate that $Zn^{2+}$ binding modulates voltage-dependent transitions in hTMEM266.

Our observations of voltage-dependent fluorescence and quencher accessibility of TAMRA attached to residues at the external ends of S3 and S4, as well as the voltage-dependent fluorescence from a GFP grafted on to the intracellular end of the S4 helix are both consistent with the conventional voltage-sensing mechanism in which the S4 helix and its positively-charged arginine residues move in response to changes in membrane voltage. However, two issues limit the mechanistic interpretation of these fluorescence measurements. First, because fluorescence is sensitive to the local fluorophore environment, a fluorescence change does not require movement of the residue to which the fluorophore is attached and can instead result from movement of other residues within its vicinity. Secondly, because hTMEM266 undergoes multiple conformational transitions, the charge movement per voltage-sensing domain cannot be reliably inferred from the voltage-dependence of the fluorescence (*Bezanilla and Villalba-Galea, 2013*). Indeed, relatively linear ΔF/F relations have been reported for a variety of genetically encoded voltage indicators engineered from the voltage-sensing domains from voltage-activated ion channels or voltage-sensitive phosphatases (*Abdelfattah et al., 2016*; *Barnett et al., 2012*; *Blunck et al., 2004*; *St-Pierre et al., 2014*; *Yang et al., 2016*). Nevertheless, our results collectively demonstrate that hTMEM266 contains a functional voltage-sensing domain that is capable of both rapid (μs) and slow (ms) structural rearrangements in response to changes in membrane voltage.

hTMEM266 has been reported to localize on the post-synaptic side of glutamatergic mossy fibers and granule cells in the cerebellum (*Kim et al., 2014*), and having established that hTMEM266 is a functional voltage sensor it will be exciting to explore whether hTMEM266 can specifically interact with intracellular signaling pathways to provide voltage-dependent regulation. In addition, the ability of hTMEM266 to respond rapidly to changes in membrane voltage offers the exciting possibility that it may be useful for designing new genetically encodable voltage indicators with particularly rapid responses. Indeed, the rapid fluorescence responses observed here are faster than what has been observed in the voltage sensors used to create the fastest available voltage indicators that are based on fluorescent proteins (*Yang and St-Pierre, 2016*; *Yang et al., 2016*).

## Materials and methods

### Bioinformatic analysis of hTMEM266 and creation of a homology model

Homologs of TMEM266 were extensively searched with BLASTp (*Altschul et al., 1990*), PSI-BLAST (https://blast.ncbi.nlm.nih.gov/) (*Altschul et al., 1997*) and HHBlits (https://toolkit.tuebingen.mpg.de/#/tools/hhblits) (*Alva et al., 2016*). Secondary and tertiary structure predictions, including the homology model of hTMEM266 were generated using the Phyre2 server (http://www.sbg.bio.ic.ac.uk/phyre2/html/page.cgi?id=index) (*Kelley et al., 2015*). The homology model was geometry

minimized using Phenix (*Adams et al., 2010*) and evaluated by MolProbity (http://molprobity.bio-chem.duke.edu/) (*Chen et al., 2010b*). Sequences orthologous to hTMEM266 were retrieved from the NCBI Refseq protein database using the BLASTp algorithm. Partial sequences or sequences of homologous VSD family members were rejected by masking the S1-S4 from the query search and specifically excluding sequences associated with keywords 'sodium, potassium, calcium, hydrogen or phosphatase' as well as restricting target lengths to between 400–800 residues in the Entrez query field. The resulting hits, most of which are predictions from automated computational analysis of genomic data, were manually curated to remove redundant sequences, incomplete isoforms as well as six sequences that were missing segments/residues of S1-S4 domains known to be essential for voltage sensing. Furthermore, only a single representative isoform, longest in length or closest in homology to hTMEM266, was chosen from each organism and multiple sequence alignment of the final set of 198 sequences was done with Clustal Omega (https://www.ebi.ac.uk/Tools/msa/clustalo/) (*Sievers et al., 2011*). The phylogenetic tree was generated by the maximum likelihood algorithm implemented in MEGA 7.0 (*Kumar et al., 2016*) (see *Figure 1—figure supplement 1*) and was visualized and illustrated with iToL (https://itol.embl.de/) (*Letunic and Bork, 2016*).

## Electrophysiological recordings of Hv1 and Kv channels

All experiments measuring proton currents were performed on HEK 293T cells (ATCC, Manassas, VA). HEK cells were not independently authenticated and mycoplasma contamination was not tested. Cells were transiently transfected using FuGene 6 (Promega, Madison, WI) with hHv1 (*Alabi et al., 2007*) under voltage-clamp using an Axopatch 200B patch-clamp amplifier (Molecular Devices, Sunnyvale, CA) and were digitized on-line using a Digidata 1321A interface board and pCLAMP 10 software (Axon Instruments, Inc). Whole-cell currents were filtered at 2 kHz using eight-pole Bessel filters and were digitized at 10 kHz. The whole-cell external solution contained (in mM): 75 NaCl, 1 EGTA, 3 $CaCl_2$, and 100 BisTris (pH 6.5) and the internal solution contained (in mM): 75 NaCl, 1 EGTA, 2 $MgCl_2$, and 100 BisTris (pH 6.5).

All Kv channel constructs were expressed in *Xenopus* oocytes. Female *Xenopus laevis* animals were housed and surgery was performed according to the guidelines of the National Institutes of Health, Office of Animal Care and Use (OACU) (Protocol Number 1253–09). Oocytes were studied following 1–4 days incubation after cRNA injection at 17°C in a solution containing (in mM) 96 NaCl, 2 KCl, 5 HEPES, 1 $MgCl_2$ and 1.8 $CaCl_2$ plus 50 µg/ml gentamycin, pH 7.6 with NaOH. Recordings were performed using the two-electrode voltage-clamp recording technique (OC-725C, Warner Instruments, Hamden, CT) with a 150 µl recording chamber. Data were filtered at 1 kHz and digitized at 5–10 kHz using Digidata 1321A interface board and pCLAMP 10 software (Molecular Devices, Sunnyvale, CA). Microelectrode resistances were 0.1–1 MΩ when filled with 3 M KCl. The external recording solution contained (in mM): 50 RbCl, 50 NaCl, 10 HEPES, pH 7.6 with RbOH at room temperature (~22°C).

## Confocal microscopy of HEK cells

One day prior to transfection HEK 293T cells were plated on poly-L-lysine coated glass coverslips. The next day cells were transiently transfected (FuGene 6) with GFP-tagged hTMEM26 and 24 hr later the cells were washed using PBS and fixed with 4% paraformaldehyde (Thermo Fisher Scientific) in PBS for 15 min. The fixed cells were then washed in PBS and permeabilized with 0.1% TritonX-100 in PBS for 10 min. F-actin was stained for 20 min using phalloidin conjugated with Alexa Fluor 647 (Thermo Fisher Scientific) that was diluted in 1% BSA-PBS. After several washes with PBS the samples were mounted onto glass slides using ProLong Diamond antifade mounting media (Thermo Fisher Scientific). Confocal images were acquired at the Neurosciences Light Imaging facility (NINDS) with an inverted laser scanning Zeiss LSM 510 microscope equipped with 488 nm Argon and 633 nm He-Ne laser lines using a 63X/1.4 Plan-Apochromat oil immersion objective. Intensity profiles were generated in Fiji (*Schindelin et al., 2012*).

## Biochemical detection of oligomerization and western blot analysis

HEK 293T cells grown in T25 25 $cm^2$ flasks were transfected with hTMEM266 and/or hHv1 cDNA using FuGene 6 (Promega). One or two days after transfection, cells were collected in Dulbecco's PBS and then centrifuged at 500 × g for 3 min. The supernatant was discarded after centrifugation,

and a lysis buffer containing 1% Triton X-100 and protease inhibitors (Roche cOmplete tablets) was used to suspend the cell pellet. Cells in lysis solution were sonicated for 5 s on ice and incubated on ice for 30 min, with vortexing for 5 s every 5 min. The cell lysate then was centrifuged at 13,000 × g for 20 min at 4°C, and the supernatant was combined with LDS buffer, DTT, and 2-mercaptoethnol. Samples were heated at 95°C for 5 min and then were centrifuged at 13,000 × g for 2 min. Proteins were separated in a 4–12% NuPage Bis-Tris gel (Invitrogen) using a running buffer containing (in mM): 50 3-(N-morpholino)propanesulfonic acid, 50 Tris base, 3.46 SDS, and 1 EDTA. SeeBlue Plus2 (Invitrogen) was used as the protein molecular-weight marker. Protein in the gel was transferred to nitrocellulose membrane and probed with rabbit polyclonal anti-hTMEM266 antibody custom raised against mTMEM266 residues 524–538.

## Crosslinking and purification of coiled coil domain

The homobifunctional crosslinker disuccinimidyl suberate (DSS, Pierce) was applied to intact HEK 293T cells expressing N-terminal GFP- or C-terminal Strep-tagged hTMEM266. The reaction was quenched with 100 mM TrisHCl (pH 8.5). Samples were collected using a Mem-PER Plus Membrane Protein Extraction Kit (Pierce) and stored at −80°C. Air oxidation samples were treated with 10 mM N-ethylmaleimide (NEM). Samples were mixed with loading dye plus reducing reagent (absent for air oxidation condition) and loaded to pre-cast, gradient 4–20% mini-PROTEAN TGX gels (Bio-Rad). Samples were transferred to nitrocellulose membrane and probed with a rabbit anti-GFP (Abcam) and anti-rabbit HRP conjugate (Abcam) or StrepMAB-Classic, HRP conjugate antibody (IBA). Western images were captured with film or digitally imaged using a ChemiDoc MP (Bio-rad) and analyzed with Image Lab (Bio-rad).

The coiled-coil domain of hTMEM266 (residues 224–289) was cloned into pET22b with an N-terminal MH$_8$SSGLVPRGS affinity tag and thrombin cleavage site and expressed in *E. coli* BL21 (DE3) cells that were induced overnight with 50 µM IPTG at 18°C. The soluble fraction of homogenized bacterial cell lysates was purified by Ni-NTA chromatography, followed by thrombin cleavage, and ion exchange chromatography at pH 8.5 on HP-Q column (GE Healthcare). Purified protein was dialyzed in to Tris buffered saline, pH 7.5 and concentrated to 8–10 mg/ml.

## Voltage-clamp fluorimetry

Mutants of hTMEM266 in the pGEM-HE vector (*Liman et al., 1992*) were constructed using polymerase chain reaction mutagenesis and verified by DNA sequencing. hTMEM266-EGFP was constructed using super ecliptic pHluorin GFP from the Ci-VSP ArcLight by replacing the C-terminus of hTMEM266 with EGFP after the S4 segment after Q233. Ci-VSP ArcLight-A242C was ordered from AddGene (Plasmid #36857), Shaker-IR with four Cys residues removed (C245V, C301S, C308S, C462A) was generously provided by Baron Chanda (Univ. Wisconsin) (*Jarecki et al., 2013*) and Ci-Hv1 was generously provided by Peter Larsson (Univ. Miami). DNA was linearized with *NheI* for hTMEM266, *SacI* for Ci-Hv1, *NotI* for Shaker, *KpnI* for Ci-VSP-Arclight242 and transcribed to RNA with the RNeasy Kit (Qiagen). *X. laevis* oocytes were injected with ~50 nL of RNA at a concentration of 0.2–1 µg/µl and incubated at 17°C for 4–12 days in an ND96 solution containing (in mM) 96 NaCl, 2 KCl, 1.8 CaCl$_2$, 1 MgCl$_2$, 5 HEPES and 50 µg/ml gentamycin, pH 7.6. Oocytes were labeled for 20–40 min. at 13°C with 10 µM TAMRA-MTS (2-((5 (6)-tetramethylrhodamine)carboxylamino)ethyl methanethiosulfonate; Toronto Research Chemicals), diluted in ND-96 without gentamycin, the extracellular recording solution used in two electrode voltage clamp fluorometry (TEVCF) for all constructs except Hv1. After labeling, oocytes were extensively washed in the recording solution and stored in the dark at 13°C prior to performing experiments. Ci-Hv1 injected oocytes were labeled with 10 µM Alexa Fluor 488 C5 Maleimide (Molecular Probes) or 10 µM TAMRA-MTS. 50 nl of 1 M HEPES (pH = 7.0) was injected into each oocyte prior to recording to minimize pH changes due to the proton efflux. The extracellular solution for Ci-Hv1 recordings contained (mM) 75 NaCl, 2 CaCl$_2$, 1 EGTA and 100 HEPES, pH 7.5. Unless otherwise indicated, chemicals were purchased from Sigma-Aldrich. Electrodes were filled with 3M KCl and had resistances between 0.1 to 1 MΩ. TEVCF recordings were obtained using a Dagan CA-1B amplifier or Oocyte Clamp OC-725C amplifier. For some experiments, cut-open vaseline gap voltage clamp fluorometry (COVCF) recordings were performed with a Dagan CA-1B amplifier (*Gandhi and Olcese, 2008*; *Rudokas et al., 2014*). The external solution for cut-open recordings contained (in mM): 105 NaOH, 20 HEPES, 2 Ca(OH)$_2$, pH 7.4

with methane sulfonic acid (MeSO$_3$H), and the internal solution contained (in mM): 105 N-methyl-D-glucamine, 20 HEPES, 2 EGTA, pH 7.4 with MeSO$_3$H. Fluorescence signals were acquired through a 40×, 0.8-NA fluorescence objective (Olympus LUMplanFLN or Nikon) on an Olympus BX51WI or Nikon Eclipse FN1 microscope and a photodiode. For TAMRA-MTS signals excitation filter, emission filter and dichroic were HQ535/50, HQ610/75 and T570pxrxt respectively (Chroma Tech.). For Alexa Fluor 488 C5 Maleimide signals excitation filter, emission filter and dichroic were ET480/40, ET535/50 and T510nm respectively (Chroma Tech.). The signal from the photodiode was low-pass filtered at 5 kHz and sampled at 20 kHz through a Digidata-1440A or a Digidata-1550 controlled by pClamp10 (Molecular Devices) and illuminated with a green LED (530 nm, M530L2-C1 from Thor-Labs) or a blue LED (470 nm, M470L2 from ThorLabs). Unless specified, fluorescence traces represent single recordings without averaging and filtered with a boxcar filter (box width 3).

## Acknowledgements

We thank Peter Larsson for the Ci-Hv1 cDNA and Baron Chanda for Shaker cDNA. We thank Lucy Forrest, Miguel Holmgren, Joseph Mindell and members of the Swartz laboratory for helpful discussions. This work was supported by the Intramural Research Program of the National Institute of Neurological Disorders and Stroke, National Institutes of Health (NIH) (KJS).

## Additional information

### Competing interests

Kenton Jon Swartz: Reviewing editor, *eLife*. The other authors declare that no competing interests exist.

### Funding

| Funder | Grant reference number | Author |
| --- | --- | --- |
| European Regional Development Fund | GINOP-2.3.2-15-2016-00015 | Ferenc Papp |
| National Institute of Neurological Disorders and Stroke | NS002945 | Kenton Jon Swartz |

The funders had no role in study design, data collection and interpretation, or the decision to submit the work for publication.

### Author contributions

Ferenc Papp, Suvendu Lomash, Orsolya Szilagyi, Conceptualization, Data curation, Formal analysis, Investigation, Visualization, Writing—original draft, Writing—review and editing; Erika Babikow, Jaime Smith, Tsg-Hui Chang, Maria Isabel Bahamonde, Conceptualization, Data curation, Formal analysis, Investigation; Gilman Ewan Stephen Toombes, Conceptualization, Formal analysis, Investigation, Visualization, Writing—original draft, Writing—review and editing; Kenton Jon Swartz, Conceptualization, Data curation, Formal analysis, Supervision, Funding acquisition, Investigation, Visualization, Writing—original draft, Project administration, Writing—review and editing

### Author ORCIDs

Gilman Ewan Stephen Toombes http://orcid.org/0000-0001-8346-1790
Kenton Jon Swartz https://orcid.org/0000-0003-3419-0765

### Ethics

Animal experimentation: All animal protocols and procedures were approved by the Animal Care and Use Committee of the National Institute of Neurological Disorders and Stroke/NIH (Animal protocol #1336).

**Decision letter and Author response**
Decision letter https://doi.org/10.7554/eLife.42372.022
Author response https://doi.org/10.7554/eLife.42372.023

## Additional files

### Supplementary files

• Transparent reporting form
DOI: https://doi.org/10.7554/eLife.42372.020

### Data availability

All data generated or analyzed is included in the figures.

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
