## [Decision Letter]

Thank you for submitting your article "TMEM266 is a functional voltage sensor regulated by extracellular Zn^2+^" for consideration by *eLife*. Your article has been reviewed by three peer reviewers, including Leon D Islas as the Reviewing Editor and Reviewer #1, and the evaluation has been overseen by Richard Aldrich as the Senior Editor. The following individual involved in review of your submission has agreed to reveal their identity: Marcel P. Goldschen-Ohm (Reviewer #2).

The reviewers have discussed the reviews with one another and the Reviewing Editor has drafted this decision to help you prepare a revised submission.

Summary:

The reviewers agree that this is an interesting and significant study that helps to clarify the function of TMEM266 as a bonafide voltage-sensing domain. The reviewers further agree that the experiments are correctly designed and carried out. However, the interpretation of the data, specially the fluorescence measurements, is not easy and is complicated by the unique, non-saturating behavior of the signals. Please see below for required revisions.

Essential revisions:

It is suggested that kinetic modeling of the fluorescence signals in the context of possible molecular mechanisms, should help clarify and provide a framework for the interpretation of experimental data.

Also, reviewers point out that if TMEM266 is a VSD, a direct evidence of its function should be obtained by measuring charge movement in the form of sensing currents. The use of the cut-open clamp used in fluorescence experiments should facilitate these experiments.

---

## [Author Response]

Essential revisions:It is suggested that kinetic modeling of the fluorescence signals in the context of possible molecular mechanisms, should help clarify and provide a framework for the interpretation of experimental data.Also, reviewers point out that if TMEM266 is a VSD, a direct evidence of its function should be obtained by measuring charge movement in the form of sensing currents. The use of the cut-open clamp used in fluorescence experiments should facilitate these experiments.

These are both excellent suggestions. We have included a paragraph in the Discussion (below) describing how the fluorescence signals fit with a conventional voltage-sensing mechanism, as well as the difficulty of uniquely assigning a mechanism from these fluorescence signals alone. We have not included a detailed kinetic analysis because the fluorescence signals can be readily described by multiple kinetic models.

The revised Discussion now reads:

“Our observations of voltage-dependent fluorescence and quencher accessibility of TAMRA attached to residues at the external ends of S3 and S4, as well as the voltage-dependent fluorescence from a GFP grafted on to the intracellular end of the S4 helix are both consistent with the conventional voltage-sensing mechanism in which the S4 helix and its positively-charged arginine residues move in response to changes in membrane voltage. […] Nevertheless, our results collectively demonstrate that hTMEM266 contains a functional voltage-sensing domain that is capable of both rapid (µs) and slow (ms) structural rearrangements in response to changes in membrane voltage.”

We have also included a paragraph in the Results section and a new supplementary figure (Figure 3—figure supplement 2) describing our efforts to measure sensing currents. We spent quite a bit of time trying to measure sensing currents, but even using the cut-open voltage clamp we could not unambiguously distinguish non-linear sensing currents from oocytes expressing hTMEM266. We did observe a significant increase in the apparent capacitance over un-injected oocytes, and we openly discuss its possible origin.

The Results section now reads:

“We also examined these current recordings for any signs of charge movement due to changes in the conformation of the protein. […] This apparent increase in membrane capacitance suggests either that hTMEM266 contains mobile charges, or that expression of hTMEM266 somehow changes the properties of the plasma membrane, such as its area, thickness or polarizability.”